# TraMEL: An Exemplar Replay-Based Continual Learning Framework for Malware Traffic Analysis

## Abstract

Most prior work on continual malware detection has focused on static code analysis. In contrast, this paper explores continual learning (CL) for malware traffic analysis (MTA), which leverages encrypted flow features to capture behavioral signals that remain observable despite obfuscation and encryption. Unlike conventional intrusion detection systems that perform coarse anomaly detection, MTA requires fine-grained family-level classification under evolving, imbalanced, and non-stationary distributions, making it a distinct and challenging setting for CL.

We introduce TraMEL (Traffic-based Malware Exemplar Learning), a replay-based CL framework designed for MTA. TraMEL integrates (i) adaptive exemplar selection to address long-tailed family distributions and (ii) an exemplar refinement phase to mitigate task recency bias under strict memory budgets. We evaluate TraMEL under both standard class-incremental and temporally shifted scenarios. Across CICAndMal2017 and IoT23, TraMEL outperforms strong CL baselines including iCaRL, ER, and TAMiL by 10–30 percentage points, and approaches the performance of joint training, a theoretical upper bound with full access to past data. These results demonstrate that CL on malware traffic is both feasible and practical, providing a memory-efficient approach toward real-world malware detection. Code is available at `https://anonymous.4open.science/r/ICLR2026-code-D575/`.

## 1 Introduction

Modern malware increasingly evades traditional defenses by encrypting network traffic (e.g., TLS 1.3) and applying code obfuscation, rendering both deep packet inspection and static analysis unreliable (Moser et al., 2007; Deng & Mirkovic, 2022; Anderson & McGrew, 2016). This shift motivates malware traffic analysis (MTA), which detects malicious activity directly from encrypted network traffic rather than executable code. Unlike conventional intrusion detection systems (IDS) that operate on coarse logs or binary anomaly flags under closed-world assumptions (Sommer & Paxson, 2010; Paya et al., 2024), MTA requires fine-grained family-level classification in an open world where malware families continually evolve and reappear (Mariconti et al., 2017). These require models that can adapt without retraining from scratch (Rahman et al., 2022). The challenge is particularly acute in mobile and embedded ecosystems, where encrypted traffic dominates and malware behavior changes rapidly. To capture this, we study two representative domains: Android malware, using the CICAndMal2017 (CIC17) dataset with 42 families (Lashkari et al., 2018), and IoT malware, using the IoT-23 dataset featuring botnets such as Mirai (Garcia et al., 2020).

Although machine learning (ML) models have achieved strong performance on static MTA benchmarks (Mirsky et al., 2018; Anderson & McGrew, 2016), we argue that this success reflects an unrealistic *closed-world assumption* (Sommer & Paxson, 2010). In real deployments, drift is driven not only by benign software evolution but also by adversary-driven evolution of malware behavior, where attackers continually release variants of known families or new malware to evade detection (Küchler et al., 2021) (see Section 2 for details). Such dynamics steadily erode classifier performance. The standard resolution, fine-tuning on new data, leads to *catastrophic forgetting (CF)*,

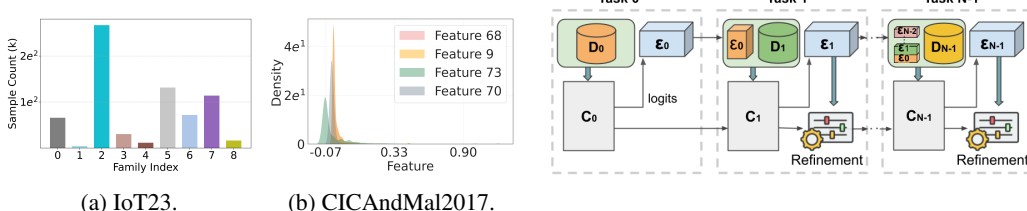

(a) IoT23.  (b) CICAndMal2017.

Figure 1: Class imbalance in IoT23 (1a) and skewed feature distribution in CIC2017 (1b).

Figure 2: Overview of the proposed TraMEL pipeline.

where the model loses the ability to detect previously learned malware families while adjusting to new threats.

A fundamental challenge is that existing malware traffic dataset do not capture long-term family recurrence, making it difficult to directly evaluate forgetting under realistic re-emergent patterns. To tackle this problem, we design two benchmarks. The first is a standard class-incremental (Class-IL) split with disjoint families, representing a strict lower bound. The second is a temporal Class-IL split grouping families by time of first appearance time to reflect natural traffic shifts. Although both lack recurrence, the temporal split is deliberately conservative, harder than real deployments where recurrence would allow transfer, and thus provides a principled benchmark for continual learning (CL) in MTA.

We therefore formalize MTA as a Class-IL continual learning problem with three objectives: (i) preserve performance on previously seen families, (ii) adapt to new families, and (iii) operate under tight memory and compute budgets. Replay-based CL is particularly well-suited here because it retains prior knowledge through compact exemplar buffers (Rahman et al., 2025). However, existing CL methods such as ER (Rolnick et al., 2019), iCaRL (Rebuffi et al., 2017a), and TAMiL (Bhat et al., 2023b) have been validated primarily in the vision domain, while CL studies in IDS settings (Channappayya et al., 2023; Amalapuram et al., 2024) focus on coarse binary anomaly detection under closed-world assumptions. Prior malware-specific CL work (Sun et al., 2025; Park et al., 2025; Rahman et al., 2025) addresses code-level drift rather than encrypted traffic. Building on these observations, we target encrypted MTA, where drift arises from both new families and re-emerging ones of older families, and mitigate catastrophic forgetting across class-incremental and temporal-drift scenarios through exemplar replay and refinement.

**Our approach.** We introduce **TraMEL** (Traffic-based Malware Exemplar Learning), an exemplar-replay CL framework tailored for MTA. TraMEL addresses three core challenges. ❶ *Long-tailed and sparse traffic features.* Real-world malware traffic exhibits long-tailed family distributions and sparse feature vectors (Figure 1). TraMEL selects exemplars that balance class coverage while preserving intra-class diversity. ❷ *Task recency bias.* Incremental training causes earlier families to be forgotten as new families are introduced. TraMEL incorporates an exemplar refinement phase that fine-tunes exclusively on buffered exemplars to reinforce prior knowledge. ❸ *Tight memory budget.* Practical malware detectors must operate with small buffers. TraMEL therefore emphasizes compact but representative exemplar selection to maintain long-term accuracy.

To this end, TraMEL combines a heuristic exemplar selection strategy—balancing class coverage with diversity-aware clustering—with an exemplar refinement phase that replays buffered samples to mitigate forgetting while maintaining adaptability.

**Results.** On CICAndMal2017 and IoT23, TraMEL consistently outperforms strong CL baselines such as iCaRL, ER, and TAMiL. Even with a buffer of only 3,000 samples (0.2% of data), it achieves about 15 percentage points higher accuracy and approaches the performance of a *joint baseline* when trained on the full dataset with access to all families at once. Clustering-based selection is especially effective under tight memory, while simpler strategies suffice when more memory is available.

## 2 THREAT MODEL

*Retrograde Malware Attack (RMA)* targets ML-based malware detectors that are incrementally updated with only *new* traffic or file samples Park et al. (2025); Rahman et al. (2025). In practice, security pipelines often retrain classifiers on fresh threat intelligence feeds (e.g., new flows, domains,

binaries) without retaining historical corpora due to storage and scalability limits. This induces *catastrophic forgetting* of earlier malware signatures and behavioral traces, enabling adversaries to weaponize legacy or lightly modified variants that evade detection. From the perspective of network traffic analysis, RMA (Rahman et al., 2025) (Park et al., 2025) unfolds in three phases (Figure 3):

- **Initial Training (①):** The detector is trained on malicious and benign traffic (e.g., packet sequences, flow metadata, TLS fingerprints).

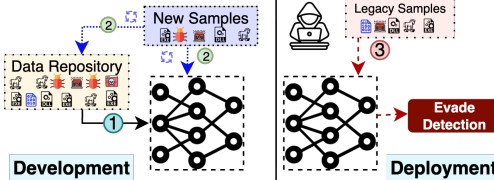

- **Updates and Forgetting (②):** The model is periodically retrained on recent captures (e.g., from honeypots or sandboxes). Because older families and domains are excluded, recall on previously seen traffic patterns declines while benign software may be misclassified, raising false positives.

Figure 3: Retrograde Malware Attack (RMA).

- **RMA in Deployment (③):** Adversaries exploit this forgetting by reintroducing legacy families or slightly altered variants.

## 3 RELATED WORK

**Replay in CL.** Addressing CF is the core challenge in CL, and one widely used solution is replay. Replay methods improve learning by mixing current data with representative information from earlier tasks. They are typically grouped into two categories – exact replay (storing real samples) and generative replay (generating synthetic data). Exact replay stores a fixed number of past samples, controlled by a memory budget $\mathcal{M}$. Methods like ER (Rolnick et al., 2019), A-GEM, and iCaRL aim to maintain performance while using as few replay samples as possible (Rolnick et al., 2019; Chaudhry et al., 2019; Rebuffi et al., 2017a). TAMiL (Bhat et al., 2023a) builds on ER by using attention to retain prior data distributions at the representation level, improving knowledge retention beyond simple replay. Generative or pseudo-replay strategies are designed to replicate the original data (Li & Hoiem, 2017; Shin et al., 2017; van de Ven et al., 2020). These techniques either generate a representative of the original data using a separate generative model or generate pseudo-data by using an earlier model's predictions as soft labels for training subsequent models.

**CL in Malware and Related Domains.** Study of CL in malware domains is relatively limited. Rahman et al.(Rahman et al., 2022) showed that replay-based methods are more effective due to the structured and diverse nature of tabular malware features. MalCL(Park et al., 2025) extends this with a GAN-based generative replay and feature-guided sampling, while MADAR (Rahman et al., 2025) introduces distribution-aware replay to select representative and discriminative samples. Beyond malware classification, other efforts address adjacent problems. Chen et al.(Chen et al., 2023) study concept drift in Android malware using contrastive and active learning, but do not tackle CF.

SPIDER (Amalapuram et al., 2024) extends CL to intrusion detection using a semi-supervised approach that matches supervised baselines while storing only unlabeled traffic. Still, it operates under a closed-world binary setting and requires up to 20% labeled data—limiting its practicality for malware. Its companion, Augmented-Memory Replay (Channappayya et al., 2023), uses only intrusion benchmarks, with limited relevance to real malware traffic and no support for privacy-preserving replay. Other work frames CL in the context of network traffic but still falls short. SPCIL (Xu et al., 2024) introduces a lightweight dual-branch model for malware detection but handles only small class increments, with growing memory and stability concerns. Zhang et al. (Zhang et al., 2025) propose an expandable CL system with per-task frozen extractors and neural architecture search. While effective on IoT and VPN datasets, these settings lack family-level malware structure and do not scale to long-horizon, evolving malware detection.

Current CL systems for network security either frame intrusion detection as a binary anomaly task or evaluate on IoT/VPN traffic, which lacks the family-level diversity characteristic of real malware. Notably, existing work does not address CL for discovering and adapting to new malware families. These limitations motivate our focus on TraMEL, which directly targets CF in the context of evolving malware families and realistic traffic streams.

---

**Algorithm 1:** TraMEL: 3 Phase Training

---

Initialize model $f$, buffer $E = \emptyset$
**for** $t = 0$ **to** $T - 1$ **do**
    **Phase 1: Initial Training**
        Train $f$ on $D_t \cup E$; save $f'$.
    **Phase 2: Exemplar Selection**
        Calculate per-class budget $m = K/C_{\text{seen}}$; truncate old exemplars to $m$ per class;
        select $m$ exemplars for each new class and update $E$.
    **Phase 3: Refinement**
    **if** $t > 0$ **then**
        Using only $E$, refine $f$ with:
        CE + distill(f, old $f$ on old exemplars) + distill(f, $f'$ on new exemplars).

**return** $f, E$

---

## 4 OVERVIEW OF TRAMEL

We present TraMEL, a continual learning framework for malware traffic classification in a class-incremental (Class-IL) setting. We assume that new malware families (i.e., classes) arrive incrementally over tasks $t_0, \ldots, t_{n-1}$, each associated with a training set $D_0, \ldots, D_{n-1}$. At task $t_i$, the objective is to train a classifier $C_i$ on the current dataset $D_i$ while retaining knowledge from previous datasets $D_0, \ldots, D_{i-1}$. In our experiments, we consider both *synthetic Class-IL splits* and more *realistic temporal shifts* assuming closed-world (i.e.,no unseen families appear in inference. In the Class-IL setup, tasks are defined by evenly partitioning malware families across $n$ tasks, which stresses the ability to recognize new families while preserving old ones. In the temporal setup, tasks are organized by the year in which malware families first appear, mimicking how new variants emerge in practice. This allows us to evaluate TraMEL under conditions where distributions evolve naturally over time, reflecting adversary-driven drift.

TraMEL addresses these scenarios through a three-phase process. First, the model is trained jointly on the current task data $D_i$ and the replay buffer $E_{<i}$ containing exemplars from earlier tasks, reducing early forgetting. Second, a set of informative exemplars is selected from $D_i$ under a fixed memory budget. The selection strategy explicitly promotes class balance and intra-class diversity, ensuring that even minority families are preserved in the buffer. Finally, to mitigate task recency bias, the model is refined exclusively on the buffer $E_i$, consolidating older knowledge without requiring full historical data.

By combining joint training, imbalance-aware exemplar selection, and targeted refinement, TraMEL achieves a balance between plasticity and stability across both synthetic and temporally defined tasks. This enables robust long-term malware detection in evolving threats. The following subsections detail the exemplar selection strategy, buffer management, and refinement procedure.

### 4.1 EXEMPLAR SELECTION STRATEGIES

Let $\mathcal{D}^c = \{(x_j, y_j)\}_{j=1}^{N_c}$ denote the training samples of class $c$, and let $f(x)$ be the feature representation of input $x$ extracted by the backbone network. The goal is to select $m$ exemplars $\mathcal{E}^c \subset \mathcal{D}^c$ for each class to be stored in the replay buffer. We investigate three strategies.

The first is random sampling, which simply draws $m$ samples uniformly from $\mathcal{D}^c$. This baseline provides unbiased coverage of the class distribution but does not exploit structure in the feature space.

The second is *class-mean selection*, following iCaRL (Rebuffi et al., 2017b). We compute the class prototype

$$\mu_c = \frac{1}{N_c} \sum_{(x_j, y_j) \in \mathcal{D}^c} f(x_j),$$

and select the $m$ samples with the highest similarity to $\mu_c$. This aligns exemplars with the class centroid, ensuring representativeness, though it may suffer from limited diversity.

The third is a *clustering-based strategy* to enhance diversity. Feature vectors $\{f(x_j)\}$ are partitioned into $k$ clusters using K-means with Euclidean distance, yielding centroids $\{\mu_{c,1}, \ldots, \mu_{c,k}\}$. Each cluster $i$ receives a quota $m_i$ proportional to its size:

$$m_i = \left\lfloor m \cdot \frac{|\mathcal{D}_i^c|}{N_c} \right\rfloor.$$

From each cluster, we select the $m_i$ samples closest to its centroid:

$$\mathcal{E}_i^c = \arg \min_{\substack{\mathcal{S} \subset \mathcal{D}_i^c \\ |\mathcal{S}| = m_i}} \sum_{x \in \mathcal{S}} \|f(x) - \mu_{c,i}\|_2^2.$$

The final exemplar set is $\mathcal{E}^c = \bigcup_{i=1}^k \mathcal{E}_i^c$. By enforcing coverage of multiple clusters, this method captures diverse semantic regions, mitigating over-representation of dense areas and improving generalization under continual learning. We empirically find that using larger numbers of clusters (e.g., $k \geq 100$) further improves performance on CICAndMal2017 and IoT23, as the buffer more faithfully reflects the underlying data manifold. Detailed results are provided in the Appendix A.2.

### 4.2 REPLAY BUFFER

Storing all past data for retraining is infeasible; instead, TraMEL maintains a fixed-size replay buffer of capacity $K$ to hold exemplars from earlier tasks. In the Class-IL setting, the number of classes grows over time while $K$ remains constant, so the quota per class decreases as tasks accumulate. If $M_i$ denotes the number of classes introduced at task $i$, then after task $i$ each class receives $\frac{K}{\sum_{j=1}^i M_j}$ exemplars. This progressive reduction makes exemplar quality increasingly critical.

Compared to vision benchmarks, where $K \leq 1{,}000$ (roughly 3% of training data) (Rebuffi et al., 2017b), malware traffic datasets require much larger buffers due to their scale. For example, maintaining the same ratio on CICAndMal2017 implies $K \approx 33{,}000$. Such scale exacerbates memory constraints and highlights the need for selection strategies that emphasize both representativeness and diversity.

To capture these practical considerations, we evaluate TraMEL under multiple buffer capacities proportionally scaled to dataset size (from 200 to 60,000), enabling a systematic analysis of how memory budgets influence exemplar effectiveness.

### 4.3 EXEMPLAR REFINEMENT

In the $i$-th task, training on the current dataset $D_i$ together with the exemplar buffer $E_{<i}$ creates a severe imbalance, since $|D_i| \gg |E_{<i}|$. This imbalance amplifies CF and leads to task recency bias, where the model favors recently observed classes (Lyu et al., 2023).

To counter this effect, TraMEL introduces a refinement phase after each task. In this phase, the model is fine-tuned exclusively on the exemplar buffer $E = E_{<i} \cup E_i$, which acts as a compact proxy for past distributions. Since exemplars are carefully selected for both representativeness and diversity, replaying them provides an efficient rehearsal step.

The refinement objective integrates supervised and distillation losses to balance plasticity and stability. Let $f^{(i)}(x)$ be the logits of the current model after refinement on task $i$, $f^{(i-1)}(x)$ the logits from the previous refined model, and $f^{(i)'}(x)$ the logits from the model immediately after task $i$ training. For an exemplar $x$ with label $y$, we define:

$$\mathcal{L}_{\text{refine}} = \mathcal{L}_{\text{CE}} + \alpha \cdot \mathcal{L}_{\text{past}} + \beta \cdot \mathcal{L}_{\text{current}},$$

where

$$\mathcal{L}_{\text{CE}} = \frac{1}{|E|} \sum_{(x,y) \in E} \text{CE}(f^{(i)}(x), y), \quad \mathcal{L}_{\text{past}} = \frac{1}{|E_{<i}|} \sum_{x \in E_{<i}} \|f^{(i)}(x) - f^{(i-1)}(x)\|_2^2,$$

$$\mathcal{L}_{\text{current}} = \frac{1}{|E_i|} \sum_{x \in E_i} \|f^{(i)}(x) - f^{(i)'}(x)\|_2^2.$$

Here, $\mathcal{L}_{\text{CE}}$ enforces correct classification across all exemplars, $\mathcal{L}_{\text{past}}$ preserves behavior on earlier tasks by aligning with the previous refined model, and $\mathcal{L}_{\text{current}}$ stabilizes adaptation to the new task by constraining deviation from the post-training model. Together, these terms mitigate recency bias while preventing overcorrection, yielding a refined balance between adaptation to emerging malware families and retention of prior knowledge. Hyperparameters $\alpha$, $\beta$, and the number of refinement epochs are scaled with buffer size and task composition; detailed sensitivity analyses are reported in Section 5.4.

### 4.4 CLASSIFIER ARCHITECTURE

Malware traffic data is inherently tabular, with each flow represented by dozens of statistical and protocol-level features rather than raw sequences or images. To identify a suitable backbone for continual learning, we evaluate three neural architectures: Multi-Layer Perceptrons (MLPs), one-dimensional Convolutional Neural Networks (CNNs), and Vision Transformers (ViTs).

The MLP baseline consists of nine fully connected layers with ELU/ReLU activations, batch normalization, and dropout. While computationally efficient, it provides limited representational power and yields the weakest performance. The CNN baseline uses a six-layer 1D convolutional stack with max pooling and fully connected layers (28M parameters). Although MLP and CNN achieve better accuracy in some settings,they suffer from instability across runs and rapid representation collapse. This reflects the limited capacity of MLP and the difficulty of applying local convolutional filters to tabular features without strong positional structure as discussed in the Appendix A.6.

In contrast, the Transformer-based model delivers both higher accuracy and greater stability. On CICAndMal2017, a ViT with six encoder blocks (hidden dimension 384, MLP size 1152, eight heads) achieves 75–80% accuracy with only 8.9M parameters. On IoT23, a lighter configuration (hidden size 16, MLP size 48, one encoder layer, two heads) achieves competitive accuracy despite the smaller input dimension. In both cases, the ViT consistently outperforms CNNs and MLPs in average accuracy and variance, while maintaining robustness across the entire Class-IL sequence.

These results provide an important insight: attention-based models are particularly well-suited for malware traffic analysis. Unlike CNNs, which rely on local receptive fields, Transformers capture global inter-feature dependencies without assuming positional priors, making them effective on tabular data where relationships among features (e.g., packet size, timing, DNS queries) are long-range and non-sequential. Moreover, the ViT achieves stronger accuracy–complexity trade-offs, with fewer parameters yet higher stability than CNNs. This aligns with recent evidence that Transformers generalize well to structured tabular data (Huang et al., 2020).

## 5 EXPERIMENTAL DETAILS

### 5.1 DATASET

We evaluate TraMEL and other replay-based CL models on two publicly available malware traffic datasets: CICAndMal2017 (Lashkari et al., 2018) and IoT23 (Garcia et al., 2020). Both datasets are split into training, validation, and test sets using an 8:1:1 ratio.

**CICAndMal2017 (1,105,290 flows).** This dataset consists of Android malware traffic spanning 42 families. During preprocessing, IP and port fields are anonymized, and traffic direction is inferred using a manually defined list of local IP addresses. Timestamps are normalized to compute inter-packet delays (IPD), which are further adjusted by traffic direction. The dataset is highly imbalanced, with the largest family containing over 75,000 flows and the smallest fewer than 4,000.

**IoT23 (712,231 flows).** This dataset contains IoT network traffic of 11 malware families. To improve class balance, we exclude two minority families (Torri and Trojan), resulting in 9 classes. Preprocessing removes timestamps, unique identifiers, host addresses, and tunneling or service-related fields. Numeric packet and byte features are log-transformed to reduce skewness. Similar to CICAndMal2017, the class distribution is highly imbalanced Hideandseek ($\sim$267k flows), Linux.Hajime (131k), and Muhstik (114k) dominate, while families like Hakai contain as few as 4,000 flows.

## 5.2 Task Configuration and Training Protocol

We evaluate two class distribution scenarios in a Class-IL setting. The first follows prior work showing that assigning more classes to the initial task can mitigate forgetting in subsequent tasks (Park et al., 2025; Rahman et al., 2022). For CICAndMal2017, we configure tasks as $M_1 = 22$ and $M_2 = M_3 = M_4 = M_5 = 5$. For IoT23, we set $M_1 = 5$ and $M_2 = M_3 = M_4 = M_5 = 1$.

The second scenario is motivated by the fact that malware families often reappear over time as new variants (Sun et al., 2025). To the best of our knowledge, no publicly available malware traffic dataset captures the same families re-emerging over time, which prevents a direct evaluation of how temporal evolution affects malware traffic analysis. Since our datasets were collected over relatively short periods, we approximate temporal dynamics by grouping malware families according to their time (year) of first appearance. For CICAndMal2017, this yields $M_1 = 4$, $M_2 = 6$, $M_3 = 6$, $M_4 = 4$, $M_5 = 10$, $M_6 = 6$, and $M_7 = 6$. The list of family names are provided in the Appendix A.10. Additional analysis using a synthetic recurrence setting is presented in Appendix A.4.

It is worth noting that in our setting, all families across tasks are disjoint. This makes the split stricter than real deployments, where families may persist and reappear, enabling transfer. Nevertheless, the overall malware-traffic distribution still shifts across tasks, so the setup remains meaningful for assessing distributional non-stationarity. Our results should be viewed as a conservative lower bound; in practice, temporal reoccurrence would likely ease the problem. Each experiment is repeated five times and we report mean accuracy; training uses 50 epochs on CICAndMal2017 and 40 on IoT23 with early stopping after the first task.

In the refinement phase, we fix a constant $k$ to balance buffer size and refinement epochs, ensuring consistent replay across settings. For CICAndMal2017, $k = 240,000$, and for IoT23, $k = 20,000$. This value is determined empirically and scales with buffer size and dataset scale.

## 5.3 Evaluation Metrics.

We report task-wise and mean accuracy as primary metrics. For each task $i$, accuracy is computed over all test samples from classes seen up to $i$, capturing both new learning and retention. Mean accuracy is the average of task-wise results, reflecting overall stability and forward transfer. To quantify CF, we use the forgetting score, defined as the per-class gap between maximum and current accuracy, which also serves to assess task recency bias.

## 5.4 Hyperparameter Tuning

We tune three key hyperparameters on CICAndMal2017 (buffer size, refinement epochs, and distillation weights ($\alpha, \beta$) and evaluate their impact using mean accuracy and forgetting score.

**Buffer size and refinement epochs.**  We vary buffer sizes between 3,000 and 33,000 and adjust refinement epochs (80 vs. 8) to keep the total number of exemplar updates per task fixed at 240K. As shown in Table 5, increasing refinement epochs effectively compensates for smaller buffers, improving retention of past knowledge.

**Distillation weights** ($\alpha, \beta$)**.**  We tune $\alpha$ to preserve past-task knowledge and $\beta$ to emphasize current-task accuracy. While $\alpha = \beta = 1$ already stabilizes learning, unbalanced settings reveal a trade-off: larger $\alpha$ improves retention but reduces new-task accuracy, whereas larger $\beta$ favors recent tasks at the cost of earlier ones. As shown in Table 5, mean accuracy remains similar across settings, but forgetting scores vary significantly, highlighting the importance of tuning ($\alpha, \beta$) for stability–plasticity balance.

## 6 Results

**Comparison to Baselines.**  We evaluate TraMEL against replay-based CL methods including iCaRL (Rebuffi et al., 2017b), ER (Rolnick et al., 2019), and TAMiL (Bhat et al., 2023b) on CICAndMal2017 and IoT23, using the same buffer size of $K = 33,000$ exemplars. For reference, we also report two standard baselines: *None*, which trains only on the current task without replay, and

Table 1: Performance of TraMEL on CICAndMal2017 (CIC17) and IoT23 datasets.

| Dataset | Model | Task 1 | Task 2 | Task 3 | Task 4 | Task 5 | Mean |
|---------|-------|--------|--------|--------|--------|--------|------|
| CIC17 | Joint | $77.12 \pm 2.5$ | $75.71 \pm 1.8$ | $76.00 \pm 1.4$ | $76.25 \pm 0.9$ | $75.61 \pm 0.3$ | $76.14 \pm 1.1$ |
| | None | $77.12 \pm 2.5$ | $14.92 \pm 2.6$ | $14.84 \pm 3.1$ | $13.37 \pm 3.6$ | $10.55 \pm 2.7$ | $26.16 \pm 1.4$ |
| | TraMEL-R | $\mathbf{76.09} \pm 1.9$ | $\mathbf{66.54} \pm 2.0$ | $\mathbf{60.36} \pm 1.4$ | $\mathbf{56.63} \pm 1.3$ | $\mathbf{53.75} \pm 0.6$ | $\mathbf{62.67} \pm 1.2$ |
| | ER | $55.23 \pm 24.3$ | $55.75 \pm 19.4$ | $59.28 \pm 4.7$ | $54.18 \pm 2.9$ | $39.60 \pm 18.7$ | $52.81 \pm 9.5$ |
| | iCaRL | $55.16 \pm 6.0$ | $29.59 \pm 17.5$ | $30.78 \pm 11.8$ | $28.31 \pm 4.8$ | $22.39 \pm 2.6$ | $33.25 \pm 6.5$ |
| | TAMiL | $57.69 \pm 15.9$ | $56.18 \pm 10.8$ | $48.79 \pm 18.5$ | $31.44 \pm 25.4$ | $47.15 \pm 7.4$ | $48.25 \pm 6.3$ |
| IoT23 | Joint | $89.55 \pm 8.6$ | $88.11 \pm 8.5$ | $85.67 \pm 5.5$ | $82.15 \pm 4.1$ | $81.99 \pm 1.0$ | $85.50 \pm 4.3$ |
| | None | $89.55 \pm 8.6$ | $23.92 \pm 24.6$ | $15.14 \pm 8.6$ | $6.69 \pm 7.1$ | $12.53 \pm 14.4$ | $29.57 \pm 7.1$ |
| | TraMEL-K | $\mathbf{89.54} \pm 9.0$ | $82.65 \pm 9.0$ | $\mathbf{76.34} \pm 10.0$ | $63.07 \pm 11.0$ | $\mathbf{59.17} \pm 18.0$ | $\mathbf{74.15} \pm 10.0$ |
| | iCaRL | $67.37 \pm 22.7$ | $67.93 \pm 18.2$ | $65.49 \pm 9.2$ | $54.51 \pm 15.2$ | $43.19 \pm 15.1$ | $59.70 \pm 7.1$ |
| | ER | $78.52 \pm 13.3$ | $\mathbf{88.21} \pm 10.9$ | $70.11 \pm 12.0$ | $\mathbf{70.17} \pm 8.8$ | $54.29 \pm 22.1$ | $72.26 \pm 2.8$ |
| | TAMiL | $81.23 \pm 18.1$ | $64.20 \pm 11.0$ | $51.06 \pm 14.8$ | $47.51 \pm 18.6$ | $52.51 \pm 15.8$ | $59.30 \pm 13.8$ |

Table 2: Performance of TraMEL on CICAndMal2017 in the temporal drift setting.

| Model | Task 1 | Task 2 | Task 3 | Task 4 | Task 5 | Task 6 | Task 7 | Mean |
|-------|--------|--------|--------|--------|--------|--------|--------|------|
| Joint | $78.48 \pm 1.7$ | $69.98 \pm 1.4$ | $66.64 \pm 0.4$ | $68.35 \pm 0.5$ | $69.44 \pm 0.4$ | $72.68 \pm 0.2$ | $73.24 \pm 0.1$ | $71.26 \pm 0.4$ |
| None | $79.73 \pm 1.9$ | $42.57 \pm 0.5$ | $39.80 \pm 0.2$ | $15.59 \pm 0.2$ | $33.62 \pm 0.1$ | $20.71 \pm 0.1$ | $13.53 \pm 0.0$ | $35.08 \pm 0.3$ |
| TraMEL-R | $79.53 \pm 1.9$ | $61.35 \pm 3.7$ | $\mathbf{53.49} \pm 2.7$ | $\mathbf{53.28} \pm 1.6$ | $\mathbf{53.07} \pm 0.6$ | $\mathbf{52.73} \pm 0.4$ | $\mathbf{50.13} \pm 0.4$ | $\mathbf{57.65} \pm 1.2$ |
| ER | $78.42 \pm 15.0$ | $\mathbf{68.22} \pm 1.4$ | $50.04 \pm 12.0$ | $48.98 \pm 17.6$ | $41.36 \pm 11.7$ | $41.87 \pm 7.7$ | $42.44 \pm 4.2$ | $53.05 \pm 2.0$ |
| iCaRL | $67.31 \pm 12.8$ | $53.90 \pm 9.6$ | $37.95 \pm 12.1$ | $28.23 \pm 9.7$ | $15.32 \pm 5.8$ | $22.09 \pm 4.6$ | $20.81 \pm 6.3$ | $35.09 \pm 6.7$ |
| TAMiL | $\mathbf{81.33} \pm 8.5$ | $55.94 \pm 15.9$ | $49.26 \pm 11.8$ | $46.51 \pm 4.6$ | $40.13 \pm 20.3$ | $38.14 \pm 8.2$ | $40.72 \pm 16.1$ | $50.29 \pm 6.2$ |

Table 3: Performance of TraMEL on the IoT23 dataset with different exemplar selection strategies.

| Method | Task1 | Task2 | Task3 | Task4 | Task5 | Mean |
|--------|-------|-------|-------|-------|-------|------|
| Random | 98.38 | 90.54 | 79.94 | **69.91** | 73.39 | 82.43 |
| C-Mean | 98.38 | 88.46 | 79.74 | 65.19 | 72.18 | 80.79 |
| KM($N_k$=600) | 98.38 | **90.78** | **80.40** | 69.28 | **75.08** | **82.78** |

*Joint*, which serves as an accuracy upper bound with full access to past and current data. Table 1 shows that TraMEL consistently outperforms iCaRL, ER, and TAMiL in both mean accuracy and stability, achieving 10–30 percentage points higher accuracy on the final task. Results in Table 2 highlight the effect of refinement: although TraMEL trails TAMiL and ER in the earliest tasks, it surpasses all baselines in later tasks under temporal drift, demonstrating stronger retention and reduced recency bias. Furthermore, to measure how quickly the method recovers and how much it retains about previously seen families, we also evaluate family recurrence in Appendix A.4.

Compared to the Joint baseline, TraMEL closes the gap by about 15 percentage points on CICAnd-Mal2017 (including the temporal shift setting) and is only about 10 points behind on IoT23. In terms of efficiency, training with $K = 33,000$ exemplars remains practical. On an NVIDIA RTX6000 Ada, the Joint requires about 6 hours, whereas TraMEL-K requires an hour, achieving competitive accuracy with significantly lower computational cost. Details are in Appendix A.3.

**Exemplar Selection.** We adopt random sampling (TraMEL-R), centroid-based selection (C-mean), K-means clustering with 600 clusters (TraMEL-K) on IoT23. TraMEL-R proves effective when memory is sufficient, while TraMEL-K provides greater robustness under tighter memory budgets by enhancing exemplar diversity. We further analyze the effect of varying the number of clusters in the Appendix A.2, which shows that larger $k$ values generally improve coverage of long-tailed distributions, peaking at around $N_k = 600$, and slightly degrades beyond that point.

**Replay Buffer Size.** To examine how buffer size influences CL performance, Table 4 reports results on CICAndMal2017 across seven capacities: $K = 200, 500, 1,000, 3,000, 6,000, 33,000,$

Table 4: Mean accuracy under varying buffer sizes on CICAndMal2017.

| Method | 200 | 500 | 1K | 3K | 6K | 33K | 60K |
|--------|------|------|------|------|------|------|------|
| TraMEL-R | 30.83 | 34.16 | 38.37 | 47.12 | 52.69 | **64.20** | **66.42** |
| TraMEL-K | 31.13 | 35.32 | 39.20 | **49.15** | **54.59** | 61.88 | 63.44 |
| iCaRL | 22.19 | 27.30 | 27.88 | 32.19 | 32.66 | 30.76 | 32.54 |
| TAMiL | 27.43 | 32.90 | 30.21 | 36.17 | 44.86 | 44.24 | 50.03 |
| ER | **35.61** | **37.15** | **39.28** | 36.53 | 44.55 | 51.42 | 57.11 |

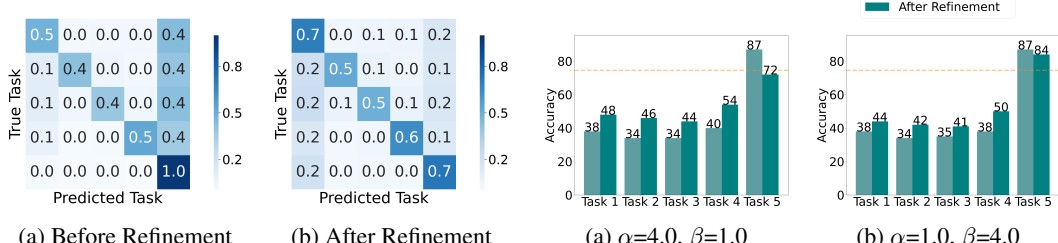

(a) Before Refinement    (b) After Refinement    (a) $\alpha$=4.0, $\beta$=1.0    (b) $\alpha$=1.0, $\beta$=4.0

Figure 4: Task-level normalized confusion matrix on CIC17 (Task 5). *Before* refinement, predictions are biased toward the current task; *after*, they are more evenly distributed, indicating reduced recency bias and forgetting.

Figure 5: Per-task accuracy on CIC17 after refinement. Larger $\alpha$ preserves past knowledge, while larger $\beta$ better maintains latest-task accuracy relative to the joint baseline.

and 60,000. All methods improve with larger buffers, but ER performs best at very small sizes ($K < 1,000$). Once the buffer reaches 1,000 exemplars, TraMEL consistently achieves higher accuracy, exceeding baselines by more than 10 percentage points when the buffer is sufficiently large to represent each class.

We also observe differences between TraMEL-R (random sampling) and TraMEL-K (K-means selection). Under tight memory budgets ($K = 200$–6,000), TraMEL-K performs better by enforcing greater exemplar diversity. As buffer size increases, random sampling becomes adequate to capture representative samples, and the performance gap between the two strategies narrows.

**Exemplar Refinement.** A key challenge in refinement is balancing adaptation to new tasks with retention of prior knowledge. This is especially critical in malware classification, where detecting newly emerging families must not come at the expense of forgetting earlier ones. Figure 4 illustrates this effect at the task level: before refinement, predictions are skewed toward Task 5, reflecting severe recency bias; after refinement, they are more evenly distributed, indicating improved stability. For detailed class-level confusion matrices, see Appendix A.1.

Figure 5 further shows how the refinement loss weights $\alpha$ and $\beta$ shape this trade-off. With $\alpha = 4, \beta = 1$ (Figure 5a), earlier tasks improve by over 10 percentage points, though Task 5 drops by 15% – still within 5% of the Joint baseline. Conversely, with $\alpha = 1, \beta = 4$ (Figure 5b), Task 5 is better preserved but forgetting of earlier tasks is more severe.

Table 5 shows that mean accuracy changes little (about 3%) across $(\alpha, \beta)$ settings under a 33,000 buffer and 8 epochs, but forgetting scores vary by up to 12% (highlighted in blue). For example, with $\alpha = 4, \beta = 1$, the forgetting scores for Tasks 2–5 are (11.74, 15.64, 18.60, 22.38), whereas with $\alpha = 1, \beta = 4$ they rise to (19.11, 28.63, 32.98, 37.41). This indicates that larger $\beta$ favors recent-task accuracy, while larger $\alpha$ better retains earlier knowledge. Hence, tuning $(\alpha, \beta)$ is essential not only for accuracy but also for managing the stability–plasticity trade-off in continual malware detection. Additionally, the refinement loss weights are explored on IoT23 (Appendix A.9), and a detailed ablation study of the refinement phase is provided in Appendix A.8.

Table 5: Mean accuracy and forgetting score across different $(\alpha, \beta)$ settings and refinement epochs, evaluated with buffer sizes of 3K (tight budget) and 33K (standard in vision benchmarks).

| Buffer | epoch | Mean Accuracy | | | Forgetting Score | | |
|---|---|---|---|---|---|---|---|
| | | $\alpha$=1,$\beta$=1 | $\alpha$=4,$\beta$=1 | $\alpha$=1,$\beta$=4 | $\alpha$=1,$\beta$=1 | $\alpha$=4,$\beta$=1 | $\alpha$=1,$\beta$=4 |
| 3,000 | 8 | 46.67 | 47.77 | 45.39 | 49.05 | 47.06 | 52.87 |
| | 80 | 50.05 | 50.75 | 48.03 | 37.09 | 32.51 | 42.51 |
| 33,000 | 8 | 59.14 | 61 | 58.44 | 27.53 | 18.02 | 30.23 |
| | 80 | 60.26 | **61.35** | 59.22 | 21.13 | **16.72** | 25.4 |

## 7 DISCUSSION

**Practicality of Memory Budget Constraints.** In practice, traffic detection systems encounter around 100K flows per sec (around billions/day) and prior NetFlow deployments report around 1.2B flows/day from a single network. As such storing all historical flows without bound is infeasible, necessitating fixed retention or sampling DN.org Staff (2025). At the same time, the malicious base-rate is tiny, deployed IDS face severe class imbalance where benign flows vastly outnumber rare attack flows (the classic base-rate problem), meaning unconstrained replay would mostly store redundant benign history Axelsson (2000).

Finally, malware-family labels arrive sparsely and expensively, creating representative labeled traffic requires specialized analyst work and multi-source correlation Guerra et al. (2022), and even "ground-truth" family datasets like MOTIF Joyce et al. (2023) needed years of threat-report curation by experts, underscoring why we cannot assume large labeled replay corpora. As such, a strict bounded memory buffer is not an artificial ML convenience but a practical abstraction of telemetry scale and labeling scarcity in continual network intrusion detection systems (NIDS) deployments.

**Limitations and Future Work.** TraMEL is designed for supervised class-incremental learning and does not leverage unlabeled samples. While semi-supervised learning is beyond the scope of this work, a simple preliminary experiment with four unlabeled classes shows that the model is less confident on unseen families than on seen ones (0.54 vs. 0.62). However, this margin is not large enough to reliably distinguish the two, suggesting that additional exploration is needed.

In this work, we adopt class-wise $K$-means exemplar selection to preserve intra-class heterogeneity, which is particularly important under long-tailed distributions where minority classes can degrade rapidly across tasks. While this strategy maintains per-class representativeness, coreset-based selection, aimed at approximating the global data distribution, has been shown to improve performance in class-imbalanced settings Mirzasoleiman et al. (2020); Hao et al. (2023). A hybrid of these approaches may therefore complement TraMEL's class-wise heterogeneity.

Another limitation is that the refinement phase introduces a trade-off that can reduce accuracy on the current task. In addition, the fixed-size replay buffer constrains scalability; as the number of classes increases, relying solely on this buffer may become less effective. Because exemplars are retained after initial selection without reselection, the buffer can drift from the evolving model, leading to a growing mismatch between stored examples and current representations. This issue could be mitigated by periodically refreshing or replacing exemplars to better align with the updated model. Extending TraMEL with an adaptive buffer mechanism is a promising future direction for improving longitudinal scalability.

## 8 CONCLUSION

We propose TraMEL, a replay-based CL framework for malware traffic analysis. TraMEL mitigates catastrophic forgetting under class imbalance and temporal shifts, yielding close to joint-training performance while operating under strict memory constraints. Nonetheless, trade-offs remain—refinement may reduce current-task accuracy, and fixed buffers limit scalability. Future work should explore scalable backbones and exemplar-free methods to better handle imbalanced distributions and consider more practical dynamic evolving samples.

REPRODUCIBILITY STATEMENT

We provide an anonymous GitHub repository containing the main source code used in our experiments. All datasets employed in this work (CICAndMal2017 and IoT-23) are publicly available, and we additionally release the preprocessing scripts to ensure consistent data preparation. The code includes a default seed, and we provide a hyperparameter as a default in the code to facilitate the reproduction of results with similar performance. In particular, seed 83, 93, 103, 113, and 123 are primarily used during the training. While we do not provide strict hardware specifications, the implementation runs without specialized dependencies and has been tested under standard GPU environments. Overall, we aim to facilitate reproducibility by releasing both the code and preprocessing pipelines, enabling independent researchers to obtain comparable results.

ETHICS STATEMENT

This study aims to classify malware families in order to strengthen defenses against malicious attacks, with no intent of misuse. No human subjects are involved, and all datasets used (CICAndMal2017, IoT-23) are publicly available.

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

# A APPENDIX

## A.1 CONFUSION MATRIX

Using the CICAndMal2017 dataset, we trained over five tasks and analyzed class-wise normalized confusion matrices before and after refinement in the last task. As shown in Figure 6a, before refinement, many samples were misclassified into the latest task. After refinement 6b, the accuracy of all classes except the latest task improved, and overly predicting to the latest task is reduced.

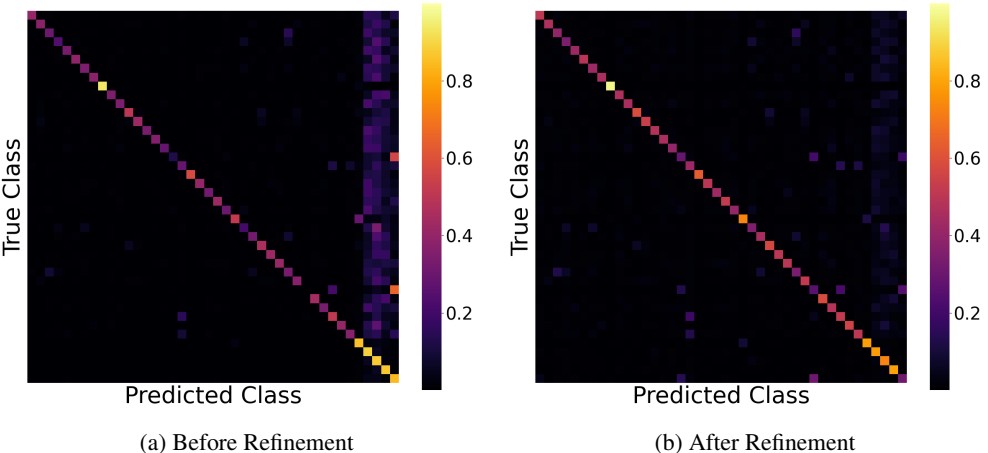

(a) Before Refinement  (b) After Refinement

Figure 6: Class-level normalized confusion matrix of before and after refinement on CIC17. Classes in recent task accuracy are over 0.8, while classes in earlier tasks have also been predicted as a last task.

Table 6: Task-wise accuracy and mean accuracy across tasks on the IoT23 dataset with different exemplar selection strategies.

| Method | Task1 | Task2 | Task3 | Task4 | Task5 | Mean |
|---|---|---|---|---|---|---|
| Random | 98.38 | 90.54 | 79.94 | 69.91 | 73.39 | 82.43 |
| C-Mean | 98.38 | 88.46 | 79.74 | 65.19 | 72.18 | 80.79 |
| KM($N_k$=5) | 98.38 | 83.00 | 72.60 | 66.60 | 71.33 | 78.38 |
| KM($N_k$=100) | 98.38 | 90.01 | 80.06 | 66.95 | 72.04 | 81.49 |
| KM($N_k$=300) | 98.38 | 88.98 | 79.71 | **70.20** | 73.03 | 82.06 |
| KM($N_k$=600) | 98.38 | **90.78** | **80.40** | 69.28 | **75.08** | **82.78** |
| KM($N_k$=800) | 98.38 | 90.68 | 79.73 | 67.88 | 72.35 | 81.60 |
| KM($N_k$=1,000) | 98.38 | 90.58 | 79.29 | 69.39 | 74.58 | 82.44 |

## A.2 CLUSTER SIZE IN EXEMPLAR SELECTION

We evaluate exemplar selection strategies, random sampling (Random), class-mean selection as used in iCaRL (Rebuffi et al., 2017b) (C-Mean), and K-means clustering-based selection (KM). Experiments are conducted on the IoT23 dataset with a fixed buffer size of $K = 10,000$, and the number of clusters $N_k$ is varied from 5 to 1,000.

As shown in Table 6, TraMEL performance improves as $N_k$ increases, peaking at around $N_k = 600$, and slightly degrades beyond that point. This trend suggests that increasing the number of clusters enhances class representation by promoting diversity in the selected exemplars. However, when $N_k$ becomes too large, the number of samples per cluster becomes too small to capture intra-class variability, reducing the representativeness of the selected exemplars. However, the optimal $N_k$ differs across datasets. IoT23 has a very small intra-class variance (about 0.08), while CIC17 is much more spread out (around 0.4), which makes the choice of $N_k$ easier to interpret. CIC17,

which has a larger variance, tends to perform better with a larger number of clusters ($N_k = 800$), whereas IoT23 reaches its best performance with a smaller value, roughly $N_k = 600$.

Overall, with $K = 10,000$, $K$-means based selection yields more representative exemplars per class than random sampling or class-mean selection, leading to better overall performance.

### A.3 COMPUTATIONAL RESOURCE

Figure 7 compares the computational cost across different buffer sizes, as well as the None and Joint baselines. We measure cost by training time on an NVIDIA RTX6000 Ada Generation GPU, with CPU parallelism limited to a single thread. The evaluation follows the same setting as Table 2. As more tasks are learned, the gap between joint training and TraMEL widens. Increasing the buffer size does not substantially raise overall cost. This is because TraMEL uses early stopping, which avoids unnecessary initial training, and even a large buffer remains much smaller than retraining on the full dataset at every task. For exemplar selection, we use the best-performing method for each buffer size, as reported in Table 4: K-means clustering-based selection (KM) for buffer sizes $K \leq 6,000$, and random selection for buffer sizes $K > 6,000$.

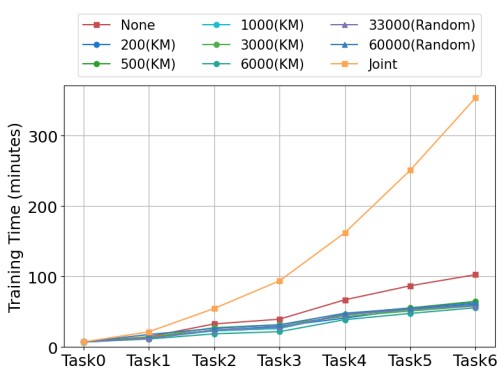

Figure 7: Computational cost of training TraMEL. Measured by training time temporal setting (Task 7) of CIC17 with seven different buffer sizes.

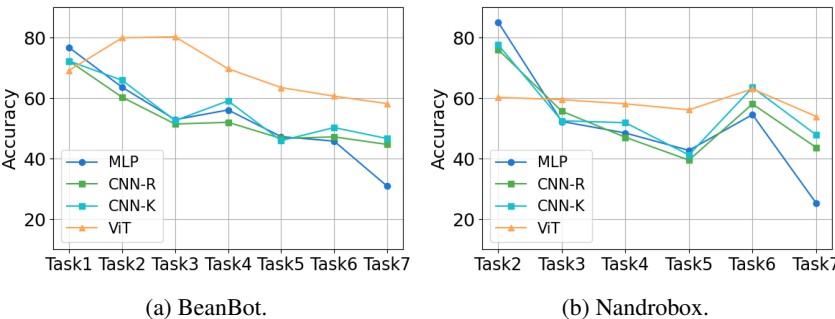

(a) BeanBot.                    (b) Nandrobox.

Figure 8: Task-level accuracy when trained family recurrence in later tasks on CIC17 (Task 7). BeanBot is initially trained in Task 1 and reappears in Task 4, while Nandrobox appears in Tasks 2 and 6.

Table 7: Performance of classifier architecture on CIC17 in the recurrence setting.

| Model | Task 1 | Task 2 | Task 3 | Task 4 | Task 5 | Task 6 | Task 7 | Mean |
|---|---|---|---|---|---|---|---|---|
| Joint | 78.41 ± 1.2 | 70.97 ± 0.4 | 66.16 ± 1.0 | 67.67 ± 1.0 | 69.28 ± 0.6 | 72.35 ± 0.3 | 72.95 ± 0.4 | 71.11 ± 0.3 |
| None | 79.15 ± 2.1 | 42.39 ± 0.3 | 39.61 ± 0.1 | 16.58 ± 0.2 | 33.42 ± 0.2 | 23.71 ± 0.2 | 13.55 ± 0.1 | 35.49 ± 0.4 |
| ViT-R | 78.42 ± 1.2 | 60.61 ± 2.1 | 55.61 ± 0.7 | **54.53** ± 0.4 | **53.59** ± 1.0 | **53.31** ± 0.6 | **50.15** ± 0.3 | **58.03** ± 0.4 |
| CNN-K | 80.87 ± 6.9 | 65.15 ± 5.6 | 52.96 ± 13.1 | 51.86 ± 1.2 | 46.92 ± 10.8 | 49.31 ± 1.0 | 44.49 ± 0.7 | 55.94 ± 3.4 |
| CNN-R | 80.87 ± 6.9 | 63.79 ± 10.5 | 53.58 ± 14.3 | 47.50 ± 16.4 | 44.47 ± 21.2 | 45.64 ± 13.4 | 41.15 ± 9.9 | 53.86 ± 10.0 |
| MLP | **83.69** ± 0.6 | **69.34** ± 0.5 | **59.03** ± 0.9 | 51.57 ± 0.5 | 50.76 ± 1.7 | 45.10 ± 1.4 | 28.12 ± 3.8 | 55.37 ± 0.7 |

### A.4 RECURRENCE OF MALWARE FAMILIES

Because CIC17 orders families temporally, it enables a natural simulation of family reappearance as new variants. To model this, we choose two families: BeanBot and Nandrobox, that first appear in

Tasks 1 and 2, respectively. Each family is split into 90% and 10%; the smaller split is treated as a new variant and inserted into Tasks 4 and 6. We use the 7-task setting and evaluate three classifier architectures: MLP, CNN, and ViT. All three models are trained with the same hyperparameter settings; the only difference is the distillation loss. MLP and CNN use KL divergence, whereas ViT uses MSE. For CNN, we additionally compare two exemplar-selection methods: K-means (CNN-K) and random selection (CNN-R).

For BeanBot (Task 1/4; Fig. 8a), ViT reaches peak accuracy around the second task and then gradually declines, while remaining consistently higher than MLP and CNN in later tasks. This is partly because Task 1 contains only four classes, allowing most BeanBot samples to remain in the buffer before truncation. Even after truncation begins, ViT preserves performance longer than the other models. A similar pattern holds for Nandrobox (Task 2/6; Fig. 8b). When BeanBot and Nandrobox reappear, both MLP and CNN recover using only 10% of samples, but this gain disappears in the following task. This behavior reflects the relative brittleness of MLP and Conv1D on the traffic dataset, whereas ViT remains more stable.

The two exemplar-selection strategies also highlight CNN's dependence on exemplar quality. In Table 7, CNN-K consistently outperforms CNN-R across most tasks. We observe that CNN accuracy fluctuates across tasks, and the refinement phase helps recover performance; however, random selection makes recovery more difficult than K-means. Overall, these results indicate that CNN is more sensitive to exemplar quality than ViT.

### A.5 F1 SCORE OF LONG-TAILED DATASET

In Figure 9, Head, Medium, and Tail groups are defined as the top 20%, middle 30%, and bottom 50% of CIC17 classes, respectively. While the Head and Medium groups decline relatively gradually across tasks, the Tail group drops much more sharply, showing that rare families are the hardest to retain.

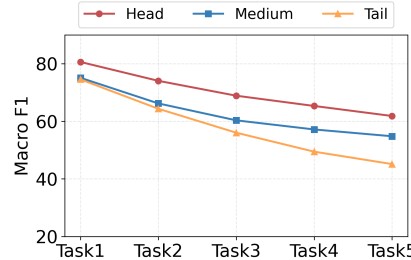

Figure 9: Macro F1 score of Head, Medium and Tail. Classes are split into 3 groups with respect to the number of samples. F1 scores are measured separately for each group.

### A.6 BACKBONE ABLATION STUDY

Backbones are evaluated under identical settings to isolate architectural effects: 5-task CIC17 split, tight buffer size 6,000, no refinement phase, and random exemplar selection in Table 8. Under these conditions, ViT shows the highest stability, achieving better accuracy and F1 scores with the lowest forgetting score. This suggests that global attention fits the malware traffic feature space better than the locality-based Conv1D or the shallow MLP. CNN performs well in the first task but suffers from large variance and a sharp drop in later tasks, while MLP consistently underperforms with a large gap between accuracy and F1. Overall, these results motivate the choice of ViT as the backbone for our framework.

Table 8: Performance of backbone on CIC17 in 5-Task.

| Model | Task 1 | Task 2 | Task 3 | Task 4 | Task 5 | Mean | Forgetting | F1 score |
|---|---|---|---|---|---|---|---|---|
| MLP | $71.36 \pm 2.4$ | $36.53 \pm 2.1$ | $28.66 \pm 3.8$ | $24.32 \pm 2.2$ | $19.17 \pm 2.0$ | $36.01 \pm 1.6$ | 64.10 | 31.31 |
| CNN | $\mathbf{78.36} \pm 1.9$ | $34.89 \pm 7.2$ | $25.57 \pm 6.3$ | $13.35 \pm 2.6$ | $11.24 \pm 3.9$ | $32.68 \pm 2.7$ | 68.67 | 28.94 |
| ViT | $74.64 \pm 2.9$ | $\mathbf{41.60} \pm 2.0$ | $\mathbf{35.72} \pm 1.8$ | $\mathbf{31.74} \pm 1.6$ | $\mathbf{29.85} \pm 1.1$ | $\mathbf{42.71} \pm 1.5$ | $\mathbf{56.04}$ | $\mathbf{42.71}$ |

### A.7 EXEMPLAR SELECTION ABLATION STUDY

In Table 9, Random, C-mean, K-means($N_k = 600$, $N_k = 800$) exemplar selection methods are presented under identical settings using a ViT encoder without the refinement phase. With ViT embeddings, C-mean fails to capture sufficiently dispersed samples in the feature space, resulting in

Table 9: Mean accuracy, F1 score. forgetting score of different exemplar selection on CIC17.

| Method | Task 1 | Task 2 | Task 3 | Task 4 | Task 5 | Mean | Forgetting | F1 score |
|--------|--------|--------|--------|--------|--------|------|-----------|----------|
| Random | $74.64 \pm 2.9$ | $41.60 \pm 2.0$ | $35.72 \pm 1.8$ | $31.74 \pm 1.6$ | $29.85 \pm 1.1$ | $42.71 \pm 1.9$ | 56.04 | 42.71 |
| C-Mean | $74.64 \pm 2.9$ | $40.48 \pm 2.7$ | $34.31 \pm 0.6$ | $31.06 \pm 1.2$ | $28.35 \pm 0.4$ | $41.77 \pm 1.4$ | 57.03 | 41.64 |
| K-Means($N_k$=600) | $74.64 \pm 2.9$ | $42.61 \pm 2.1$ | $37.85 \pm 2.1$ | $33.79 \pm 1.2$ | $31.60 \pm 1.5$ | $44.10 \pm 1.4$ | 53.55 | 44.49 |
| K-Means($N_k$=800) | $74.64 \pm 2.9$ | $\mathbf{43.43 \pm 2.7}$ | $\mathbf{37.96 \pm 2.0}$ | $\mathbf{35.21 \pm 1.4}$ | $\mathbf{32.44 \pm 1.3}$ | $\mathbf{44.74 \pm 1.5}$ | **52.71** | **45.24** |

low diversity. In contrast, Random selection, by sampling uniformly at random, captures a diverse set of samples and performs better than C-mean. K-means with both cluster sizes ($N_k = 600, 800$) outperforms these methods by selecting well-spread samples. In detail, K-means with $N_k = 800$ achieves even better performance than $N_k = 600$ in this regard.

Table 10: Comparison of three refinement phase loss settings: no refinement, (i) CE-only refinement ($\alpha = 0, \beta = 0$), (ii) CE with distillation refinement ($\alpha = 4, \beta = 1$), and (iii) Distillation-only refinement.

| Method | Task 1 | Task 2 | Task 3 | Task 4 | Task 5 | Mean | Forgetting | F1 score |
|--------|--------|--------|--------|--------|--------|------|-----------|----------|
| No-Refinement | $74.64 \pm 2.9$ | $43.43 \pm 2.7$ | $37.96 \pm 2.0$ | $35.21 \pm 1.4$ | $32.44 \pm 1.3$ | $44.74 \pm 1.5$ | 52.71 | 45.24 |
| (i) CE-only | $74.64 \pm 2.9$ | $55.03 \pm 2.6$ | $49.02 \pm 2.0$ | $45.45 \pm 0.8$ | $41.44 \pm 0.6$ | $53.12 \pm 1.5$ | 31.95 | 51.24 |
| (ii) CE with Distillation | $74.64 \pm 2.9$ | $59.31 \pm 1.7$ | $51.45 \pm 1.6$ | $46.58 \pm 1.0$ | $\mathbf{42.45 \pm 0.7}$ | $54.89 \pm 1.5$ | 25.97 | **52.70** |
| (iii) Distillation-only | $74.64 \pm 2.9$ | $\mathbf{59.37 \pm 1.9}$ | $\mathbf{51.56 \pm 1.6}$ | $\mathbf{46.68 \pm 0.9}$ | $42.23 \pm 0.8$ | $\mathbf{54.90 \pm 1.5}$ | 25.78 | 52.61 |

## A.8 REFINEMENT PHASE ABLATION STUDY

In Table 10, this study evaluates three refinement methods compared to a no-refinement baseline. Under the same experimental setting (ViT backbone, buffer size of $6,000$, K-means with $N_k = 800$), we compare: (i) CE-only Refinement($\alpha = 0$, $\beta = 0$), (ii) CE with Distillation Refinement($\alpha = 4$, $\beta = 1$), (iii) Distillation-only Refinement(CE weight=0, $\alpha = 4$, $\beta = 1$).

All three methods outperform the no-refinement baseline. Method (ii) achieves higher accuracy than (i), showing the benefit of combining CE and distillation. Method (iii) yields slightly higher accuracy than (ii) because it weighs more on past knowledge, resulting in a lower forgetting score. However, since $\alpha$ and $\beta$ can be tuned empirically, to maintain the balanced performance across tasks, removing CE is suboptimal. In particular, method (iii) exhibits a lower macro-F1 than (ii), indicating that CE is important for maintaining balanced performance across tasks.

## A.9 REFINEMENT LOSS ON IOT23

Table 11: Mean accuracy, forgetting score and F1 score across different $(\alpha, \beta)$ on IoT23 under 10K buffer with k-means selection.

| | $\alpha$=0,$\beta$=0 | $\alpha$=1,$\beta$=1 | $\alpha$=4,$\beta$=1 | $\alpha$=1,$\beta$=4 | $\alpha$=4,$\beta$=4 |
|--------|--------|--------|--------|--------|--------|
| Accuracy | 76.20 | 76.30 | **76.76** | 75.88 | 76.32 |
| F1 Score | **70.62** | 69.09 | 69.34 | 68.82 | 68.97 |
| Forgetting Score | 17.82 | 13.94 | **13.76** | 14.58 | 13.66 |

Table 11 reports the effect of the refinement loss weights $\alpha$ and $\beta$ on IoT23. This experiment uses 20 refinement epochs with a 10K replay buffer. When $\alpha$ (weighing past) is larger than $\beta$, the model achieves the highest overall accuracy along with the lowest forgetting score. We also evaluate the case without any distillation loss ($\alpha = 0$, $\beta = 0$). Interestingly, on IoT23, removing distillation still yields competitive accuracy, while forgetting score is the lowest among all. This suggests that the distillation loss plays an important role in mitigating task-recency bias by stabilizing previously learned representations.

Table 12: 7-Task temporal split of CIC17.

| Task | Families |
|------|----------|
| M1 | BeanBot, Plankton, SMSsniffer, Zsone |
| M2 | Penetho, Biige, FakeMart, FakeNotify, Jifake, Nandrobox |
| M3 | AndroidDefender, AVpass, FakeAV, FakeJobOffer, FakeTaoBao, FakeInst |
| M4 | Selfmite, Pletor, Svpeng, VirusShield |
| M5 | Kemoge, Mobidash, Shuanet, Youmi, Koler, LockerPin, Simplocker, AV for Android, FakeApp, FakeApp.AL |
| M6 | Dowgin, Feiwo, Gooligan, PornDroid, AndroidSpy.277, Mazarbot |
| M7 | Ewind, Koodous, Charger, Jisut, RansomBO, WannaLocker |

## A.10 CIC17: 7-TASK TEMPORAL SPLIT

Table 12 presents the temporal split of families in CIC17, which follows the year-based grouping used in Lashkari et al. (2018). Each task contains the malware families that emerged in a specific year between 2011 and 2017.

