# OpenReview forum: "TraMEL: An Exemplar Replay-Based Continual Learning Framework for Malware Traffic Analysis"
_ICLR.cc/2026/Conference — Submitted to ICLR 2026_

### Official Review · Reviewer_jRDc · 2025-10-25

**Soundness:** 3
**Presentation:** 2
**Contribution:** 2
**Rating:** 2
**Confidence:** 3

**Summary:**

This paper proposes a continual learning framework for malware traffic analysis, namely TraMEL.
The methodology relies on large replay buffer, and the training loss which penalizes regression of accuracy on previously-correctly-classified samples. The classifier is then chosen among different strategies, selecting transformers for this cause.
When compared with the state of the art of continual learning (not continual learning dedicated to traffic), results are mixed, with ER competing with TraMEL (even if ER is not strictly for malware detection).

**Strengths:**

+ this paper ships together interesting techniques to provide a continual learning framework for traffic malware detection
+ interesting choice of including a regression-free loss function into the objective function
+ it is also interesting that transformers are acting well on tabular data

**Weaknesses:**

The paper is interesting, but the scope is very narrow for being considered by a venue like ICLR.
This contribution is more suited to a Security conference (A or B) like EuroS&P, ECML, SAC, or workshops like AISec, DLS, WoRMA, etc.
The reasons behind these comments are the following:

**1) Incremental contribution that does not provide insights on how to use domain knowledge.** The main modification is using the regression-free loss, while all the other components seem to be already investigated and proposed by prior work, outside the malware domain. Also, while the paper clearly states at the beginning that domain knowledge is rarely used in these contexts, it falls into the same problem. If the issue was the length of the replay buffer (which the experiments show it was) then also other methods outside security can be used by tweaking that parameter. Also, the strict resource requirements are mentioned, but the paper does not provide a quantitative way to judge such claim. Thus, the contribution feels more as an application of a new loss rather then providing systemic insights in the domain.

**2) Not discussed why there is a boost in performance.** Is it because the transformer is useful? Or is the sampling? Only the buffer replay size is discussed, but no complete ablation study is provided.

**3) Tabular data = ensembles of trees.** Why the paper deploys deep networks, when tabular data is empirically proven to be the best data format for ensembles of trees? See attached references. However, this would not be a contribution, but the correct usage of prior works published on top-tier venues.

**4) No limitations are discussed.** The paper has limitations, related to the choice of architecture, of the choice of the dataset, and other many that the paper does not take into account, being a huge miss for this paper.

**5) Missing baselines.** While the paper states that there are few work on continual learning for traffic data, some are presented. It is not clear why they were not evaluated. The papers state they are just not practical, without being convincing on why.

**Minor comments.**
1. there is an Appendix ?? reference broken at page 4
2. space is used strangely, with paragraphs with different gaps
3. RMA attacks exist? Is there a reference to those? Or is it a novel concept here?
4. Figure 1 is not really informative for the paper.
**References.**

Grinsztajn, L., Oyallon, E., & Varoquaux, G. (2022). Why do tree-based models still outperform deep learning on typical tabular data?. Advances in neural information processing systems, 35, 507-520.

Shwartz-Ziv, R., & Armon, A. (2022). Tabular data: Deep learning is not all you need. Information Fusion, 81, 84-90.

Gorishniy, Y., Rubachev, I., Khrulkov, V., & Babenko, A. (2021). Revisiting deep learning models for tabular data. Advances in neural information processing systems, 34, 18932-18943.

Fernández-Delgado, M., Cernadas, E., Barro, S., & Amorim, D. (2014). Do we need hundreds of classifiers to solve real-world classification problems?. The journal of machine learning research, 15(1), 3133-3181.

**Questions:**

1) Which is the reason that bring a boost i performance? The architecture of the classifier? The replay buffer? The regression-free loss? Or is it the combination?
2) Why the paper uses complex networks, where ensembles of trees are empirically proved the best on tabular data (given a good feature extractor)?
3) What is exactly the contribution w.r.t. standard continual learning methods?

---

> ### Author Response · Authors · 2025-12-02
>
> Thank you for your valuable feedback. We address the concerns as follows:
>
> **1. Contribution**
>
> We respectfully disagree with the reviewer. Based on our experience in machine learning and security, we argue that simple ``tweaks” are not sufficient in this domain. In practice, systems that rely on ad-hoc adjustments are rarely considered reliable enough for deployment in industry. To support this point, we conducted extensive evaluations to identify which factors actually drive continual learning performance on encrypted malware traffic. We also added a discussion in the revised manuscript to clarify the practical implications and address the domain specific confusion.
>
> **2. Boost in Performance**
>
> To empirically show how the model yields performance, we have performed an extensive ablation studies of the backbone, exemplar selection, and the refinement loss in Appendix A.6, A.7, and  A.8.
>
> Appendix A.6 shows that the ViT encoder produces more stable and resilient representations than Conv1d  or MLP. While CNN fluctuates across tasks, MLP underperforms consistently, whereas ViT maintains strong performance across tasks.
> This distinction is crucial because both K-means exemplar selection and the refinement phase rely on the quality of the embedding space. ViT produces more dispersed embeddings, enabling K-means with a larger cluster size $N_k$ to select exemplars that better cover the class distribution than other exemplar selection, as shown in Appendix A.7.  These well-distributed exemplars enhance the effectiveness of the refinement phase. Even without distillation loss ($\alpha$, $\beta$ = 0), the model remains competitive; however, using ($\alpha$ = 4, $\beta$ = 1), performance improves further by mitigating task recency bias.
>
> **3. Why Using Deep learning Network**
>
> Although tree-based models are stronger on standard tabular data, our setting is not static classification but constrained buffer CL. In our experiments, the Random Forest model achieved 87% on CICAndMal2017, and 99% on IoT23 in the first task of 7 task settings (Table 2). However, the performance decreases when rehearsal with a replay buffer. Unlike storing all data, under a replay rehearsal, its performance collapses quickly.
>
> This limitation is especially evident in the recurring setting, where Beanbot and Nandrobox are split into 90% (initial) and 10% (reappear). Even under a joint training setting, Beanbot accuracy fell from 0.8690→ 0.8075 → 0.2659 → 0.3029 → 0.2831 → 0.2831 → 0.3069, and Nandrobox from 0.8075→ 0.7428 → 0.7171 → 0.6893 → 0.6897 → 0.6686. The difference between the two reflects the sample imbalance: the number of Nandrobox samples is roughly 6 times that of Beanbot. These results show that the tree ensemble is sensitive to data imbalance and cannot recover old decision boundaries from small recurrence samples. In contrast, ViT-based embeddings remain stable with a constrained buffer.
>
> **4. Limitations**
>
> A key limitation of this work is its dependence on a fixed-size replay buffer. Because the buffer size is constrained, scalability becomes challenging when the number of tasks is infinite. Also, keeping samples after selecting exemplars may lead to a mismatch between the stored exemplars and the evolving model. Additional discussion of these limitations is provided in the revised version's Discussion section.
>
> **5. Missing Baselines**
>
> We selected replay-based baselines because they are domain-agnostic and operate directly on flow-level metadata, which is the setting of malware traffic analysis. Prior CL works in network security (e.g., SPIDER) are semi-supervised or anomaly-detection frameworks, and are therefore incompatible with our supervised family classification task. To strengthen the comparison, we attempted to include the coreset-based replay method, which is expected to handle class imbalance well. However, we were unable to reproduce the implementation due to issues with reproducibility in the released codebase. For clarity, we have added an additional explanation in the Discussion section that contextualizes the differences between our approach and prior CL methods.
>
> **6. Minor comments**
>
> 1 / 2. We corrected the minor comments issues.
>
> 3. RMA Attack is referenced from MADAR (Rahman et al., CAMLIS 2025), and MalCL (Park et al., AAAI 2025).
>
> 4. Figure 1 is important because it presents the problem statement by illustrating characteristics of the datasets: long-tailed class imbalance and feature skewness. These properties explain why standard CL methods struggle in this setting and why exemplar selection and refinement are required.

---

### Official Review · Reviewer_XFd3 · 2025-10-29

**Soundness:** 3
**Presentation:** 2
**Contribution:** 2
**Rating:** 4
**Confidence:** 4

**Summary:**

This paper presents the TraMEL framework for fine-grained malware family classification within non-stationary network traffic. The framework addresses the challenges of catastrophic forgetting and class imbalance through a memory-efficient strategy that combines diversity-aware exemplar selection and an exclusive exemplar refinement phase.

To achieve this, it introduces a dual-loss objective during refinement to explicitly counteract task recency bias, thereby maintaining a robust balance between stability and plasticity. Empirically evaluated on two datasets under both class-incremental and temporal-shift scenarios, it demonstrates performance gains over strong replay-based baselines. This closely approaches the theoretical upper bound of joint training accuracy.

**Strengths:**

+ The paper formalizes malware traffic analysis as a unique and challenging Class-IL problem.

+ The introduction of an exclusive refinement phase, combined with tailored distillation losses ($\mathcal{L}_{past}$ and $\mathcal{L}_{current}$), is a well-motivated and empirically demonstrated method for countering the prevalent task recency bias.

+ The finding that transformer architectures are superior for malware traffic's tabular data structure is important.

+ The framework achieves significant and consistent performance increases (10-30 percent) over strong, established CL baselines across multiple datasets.

**Weaknesses:**

- Unclear use of some terms.

- The selection of datasets and baselines should be argued better.

- The refinement mechanism and fixed buffer size inherently limit scalability, as the efficacy of rehearsal degrades dramatically if the memory budget cannot be proportionally maintained as the number of tasks grows indefinitely.

- The cluster size ($N_{k}$) for exemplar selection and the distillation weights ($\alpha, \beta$) require extensive empirical tuning, which may make it complex to deploy or generalize to new, unseen datasets without a similar tuning effort.

**Questions:**

The following needs clarification, better arguments to make the paper stronger.

(1) The exemplar refinement phase is great, but I felt that its computational burden and dependency on specific loss weight tuning deserve a deeper discussion regarding future scalability. Specifically, while the refinement effectively counters recency bias, it introduces a separate, mandatory optimization step after every task, increasing the wall-clock time and computational load relative to standard replay methods. The paper can consider quantifying the exact overhead (e.g., training time in minutes/hours) added by the refinement phase compared to the joint training phase alone, especially in the tightest budget scenarios.

(2) The K-means clustering-based selection (TraMEL-K) is shown to be crucial for tight memory budgets, but its reliance on a pre-determined optimal cluster number ($N_{k}=600$ for IoT23) is unclear to me. Here, I understand that the dependence of performance on the specific, empirically found value of $N_{k}$ (Section A.2, Table 6) suggests that this hyperparameter may not generalize well. If $N_{k}$ must be tuned per dataset, it negates some of the framework's practical use cases.


(3) The paper defines the temporal-shift scenario as disjoint families grouped by the year of first appearance, which is acknowledged as a stricter than real deployments lower bound. Although the retrograde malware attack threat model is excellent, the current temporal-shift benchmark does not fully capture the recurring nature of the RMA's third phase (reintroducing legacy families or slightly altered variants). This is because the paper uses only disjoint families, and it assesses drift, not genuine recurrence.

(4) The paper's foundational arguments on buffer constraints, related work, and core assumptions need stronger substantiation to clearly frame its novelty.

- The paper repeatedly asserts that detectors must operate under strict memory budgets due to storage and scalability limits. This motivation is currently asserted without concrete, domain-specific justification.

-  The term closed-world assumption is central to differentiating this work from traditional IDS, but it is never formally defined.

-  The Related Work section is dismissive toward several recent, relevant CL efforts in network security, primarily by stating their limited relevance.  I believe,  instead of dismissing related CL works (\eg SPIDER, SPCIL) or other malware CL methods (\eg MalCL), the paper can briefly and precisely articulate the technical difference in feature space.

---

> ### Author Response · Authors · 2025-12-02
>
> Thank you for your valuable feedback. We address the concerns as follows:
>
> **1. Use of Terms**
>
> We have clarified some terms such as closed-world in the revised version. In our setting, the closed-world refers to all the evaluated malware families that are trained in the training, which follows standard Class-IL with the goal of mitigating forgetting across known families. While open-world evaluation of unseen families is important, our focus is on catastrophic forgetting within the known family under strict memory constraints.
>
> **2. Selection of Dataset and Baseline**
>
> Modern malware frequently communicates over encrypted channels such as TLS 1.3 and QUIC, which makes traffic analysis considerably more challenging. Prior works [1], [2] have explored transforming raw traffic into images to capture patterns. However, such approaches are computationally expensive, particularly in a continual learning setting. For this reason, a flow-level dataset that provides a compact and informative representation of encrypted traffic is practical; we chose this dataset.
>
> Furthermore, the characteristics of malware dataset influence the choice of baselines. Unlike image data, Convolutional neural networks achieve high accuracy but often show instability on malware traffic data. In contrast, replay-based methods are domain-agnostic and known for their applicability across diverse domains among continual learning baselines. Also, the goal of TraMEL is to improve replay-based method; we focused only on replay baselines in this paper.
>
> [1] "Explainable Artificial Intelligence for Improving a Session-Based Malware Traffic Classification with Deep Learning." 2023
> [2] "A generalizable approach for network flow image representation for deep learning." 2023
>
> **3.  Scalability Limitation**
>
> We would like to clarify that as the number of tasks increases, the buffer inevitably stores older samples, which can drift out of sync with the updated model and lead to a growing mismatch between the model and these outdated exemplars. This imitation, however, is inherent to all Replay-based CL methods, becoming more pronounced under infinite tasks. In our discussion section, we add this issue to consider periodically refreshing or replacing exemplars with more representative  samples as the model is updated. Extending TraMEL toward an adaptive buffer mechanism would be a useful future direction for improving longitudinal scalability.
>
> **4. Empirical Tuning**
>
> We agree that empirical tuning is required to achieve the best performance. However, TraMEL is not critically dependent on precise tuning for practical deployment:
>
> (1) Distillation loss weights ($\alpha$, $\beta$)
>
> As shown in the Appendix A.8, even without the distillation loss ($\alpha$, $\beta$ = 0), the accuracy remains competitive. The weights $\alpha$ and $\beta$ modulate how strongly the model preserves past knowledge vs. recent task knowledge. In this regard, $\alpha$ = 4 and $\beta$ = 1 work robustly across both CICAndMAl2017 and IoT23.
>
> (2) Cluster size $N_k$ for K-means exemplar selection
>
> We observed that larger $N_k$ effectively captures the distribution of ViT encoder embeddings. Thus, beyond a moderate range, performance becomes less sensitive to the exact value of $N_k$, as shown in the appendix A.7. However, because the datasets have different embedding distributions and intra-class variance, the optimal $N_k$ can differ. To offer a practical guideline, we recommend choosing $N_k$ based on intra-class variance: classes with higher variance benefit from larger $N_k$. For example, the intra-class variance of CICAndMAl2017 ($N_k$=800) is 0.4, and that of IoT23 ($N_k$=600) is 0.08.
>
> **5. Computational Overhead**
>
> We agree that adding the refinement phase introduces additional computational overhead. This was briefly acknowledged in the original submission, but we have now expanded the discussion in detail in Appendix A.3. While the previous results were based on a 5-task setting, we additionally conducted a 7-task setting experiment to more clearly illustrate the cumulative computational burden. We also include different replay buffer budgets, along with None and Joint baselines in Figure 7.
>
> **6. Recurrence Experiment**
>
> To address this concern, we added a recurrence experiment in which two classes in early tasks (Task  1/2) reappeared in the later tasks (Task 4/6). Beanbot and Nandrobox were split into 10% and 90%, and 10% of the dataset is used for the later tasks. This setup evaluates recurrence rather than temporal shift and reflects the re-emergence pattern described in retrograde malware attack threat model. Also, we include different feature extractor to demonstrate how ViT is well-suited to handling RMA. Details are attached in Table 7 and Figure 8 in the Appendix A.4.

---

> > ### Author Response · Authors · 2025-12-02
> >
> > **7. Memory Budget Constraints**
> >
> > In practice, traffic detection systems encounter around 100K flows per sec (~billions/day) and prior NetFlow deployments report ~1.2B flows/day from a single network. As such storing all historical flows without bound is infeasible, necessitating fixed retention or sampling [1].
> > At the same time, the malicious base-rate is tiny, deployed IDS face severe class imbalance where benign flows vastly outnumber rare attack flows (the classic base-rate problem), meaning unconstrained replay would mostly store redundant benign history [2].
> >
> > Finally, malware-family labels arrive sparsely and expensively, creating representative labeled traffic requires specialized analyst work and multi-source correlation [3], and even ``ground-truth” family datasets like MOTIF [4] needed years of threat-report curation by experts, underscoring why we cannot assume large labeled replay corpora.
> > As such, a strict bounded memory buffer is not an artificial ML convenience but a practical abstraction of telemetry scale and labeling scarcity in continual NIDS deployments.
> >
> > [1] https://dn.org/correlating-dns-and-netflow-in-a-unified-big-data-platform-for-comprehensive-network-visibility
> >
> > [2]  Axelsson, Stefan. "The base-rate fallacy and the difficulty of intrusion detection." ACM Transactions on Information and System Security (TISSEC) 2000.
> >
> > [3] Guerra, Jorge Luis, Carlos Catania, and Eduardo Veas. "Datasets are not enough: Challenges in labeling network traffic." Computers & Security 2022.
> >
> > [4] Joyce, Robert J., et al. "Motif: A malware reference dataset with ground truth family labels." Computers & Security 2023.
> >
> > **8. Related Work**
> >
> > While SPIDER, SPCIL, and MalCL are all continual-learning approaches in the security domain, they operate on different feature spaces and problem formulations. For example, SPIDER formulates continual learning as semi-supervised network intrusion detection, which detects benign and malicious flows. SPCIL detects malware categories using supervised learning by transforming PCAP data into an image, while MalCL proposes a GAN-based replay framework for Class-IL malware family classification on static features extracted from Windows PE and Android APK. In contrast, our work formulates CL as supervised malware family classification using flow-level metadata extracted from encrypted traffic, along with realistic settings such as family recurrence.

---

### Official Review · Reviewer_2rZ4 · 2025-10-30

**Soundness:** 3
**Presentation:** 3
**Contribution:** 2
**Rating:** 4
**Confidence:** 3

**Summary:**

This paper proposes TraMEL, an exemplar replay-based continual learning (CL) framework for fine-grained malware family classification from encrypted network traffic. It addresses catastrophic forgetting in class-incremental (Class-IL) and temporal shift scenarios, evaluating multiple exemplar selection strategies—random sampling, class-mean (following iCaRL), and clustering-based (K-means) to handle long-tailed distributions, along with fixed-memory buffer management and a refinement phase using distillation losses weighted by hyperparameters α and β to mitigate recency bias. Evaluated on CICAndMal2017 (Android, 42 families) and IoT23 (IoT, 9 families) datasets using disjoint class and temporal splits, TraMEL outperforms baselines like ER, iCaRL, and TAMiL by 10-30% in accuracy, approaching joint training under tight buffers (0.5% of data, 3000 samples). Additional contributions include the Retrograde Malware Attack (RMA) threat model and exploration of backbones (MLP, CNN, ViT).

**Strengths:**

Novel problem framing: TraMEL extends continual learning to malware-traffic analysis, addressing encrypted flow data rather than static code features an underexplored yet operationally realistic domain.

Refinement for bias mitigation: The two-phase training scheme, joint training followed by exemplar-only refinement, substantially reduces recency bias.

Interpretable stability–plasticity control: The analysis of the refinement loss coefficients (α, β) offers clear insight into how past-knowledge preservation and current-task retention interact, enabling tunable control over forgetting.

The evaluation in two complementary settings: standard class-incremental learning (Class-IL) with disjoint families as a strict worst-case benchmark and a temporal split grouping families by time of first appearance to simulate natural drifts, offers a thorough assessment of the model's robustness, capturing both controlled and realistic evolutionary dynamics in malware traffic.

Memory-efficient: Fixed-capacity buffers compared to baselines with proportional per-class quotas scale from 200–60 k exemplars, maintaining near-joint accuracy while operating within realistic storage limits

**Weaknesses:**

Lack of imbalance-aware evaluation: The accuracy results give a solid picture of overall
performance, showing the thorough testing done. For even better understanding of how the
model deals with rare malware types where common ones might overshadow them, adding
balanced metrics that weigh all types equally (like average F1-score, recall across classes, or
precision-recall curves per class) would be helpful.

Lack of open-world/label-scarce robustness. Testing with fully labeled and known malware
types sets a strong foundation. To make it more applicable to real-world situations where new
threats show up without labels, adding checks for handling unknown attacks or learning from
partially labeled data would be a nice step. An ablation study on completely unknown malware
types can be helpful to determine the real-world scenario

Cross-dataset tuning fairness is unclear. The tuning of key settings (α, β) on one dataset
(CICAndMal2017) is clearly described and thoughtfully done. To build more trust in results
across datasets, noting if the other dataset (IoT23) used the same settings or was tuned on its
own would ensure everything is fair and make it easier for others to repeat or expand the work.

Backbone ablation is underreported. Using a Vision Transformer as the main model looks
promising, with the initial results suggesting it beats simpler options like multi-layer perceptrons
or convolutional networks. To back this up fully and give a clearer view, adding more detailed
numbers for those alternatives in the comparison section would make the advantages stand out
better.

**Questions:**

The paper excludes two minority families (Torri, Trojan), reducing IoT23 from 11→9
classes. Could you clarify the rationale? This would provide valuable context for readers
replicating or extending your experiments.

When 𝐾 is full and per-class quotas shrink after task 𝑖, which exemplars are removed
from earlier classes, and do you re-encode/re-cluster old classes under the updated
model?

Were MLP, CNN, and ViT evaluated under identical settings, like the same buffer 𝐾,
refinement budget 𝑘, and distillation weights (𝛼, 𝛽) when the claim about ViT’s superior
performance was made?

---

> ### Author Response · Authors · 2025-12-02
> **Rebuttal Response to Reviewer 2rZ4**
>
> Thank you for your valuable feedback. We address your concerns as follows:
>
> **1. Imbalance-aware evaluation**
>
> We split the CICAndMal2017 families into three groups based on their sample counts: Head (top 20%), Medium (middle 30%), and Tail (bottom 50%). The mean of F1 score for each group across tasks is reported in Figure 9 in the Appendix A.5 in the revised version. In Figure 9, Head and Medium exhibit a gradual decline, whereas Tail classes drop more rapidly due to their low sample frequency.
>
>
> **2. Open-world/label-scarce**
>
> Although TraMEL focuses on fully supervised class-incremental learning, we conducted a small preliminary experiment to examine the model’s confidence in unknown samples. Specifically, we measured the mean maximum softmax scores for four unlabeled (unseen) families and thirty-eight labeled (seen) families in CICAndMal2017. We observe that the model is more confident on seen samples than unseen ones (0.62 vs. 0.54), but the gap was not sufficiently large, indicating that the current model does not inherently distinguish between labeled and unlabeled samples. We agree that handling unknown or partially labeled malware is an important direction for future work. We discuss this in the discussion section in the revised version.
>
> **3. Hyperparameters (α, β)**
>
> We used the same hyperparameter settings (α = 4, β = 1) for both CICAndMal2017 and IoT23. These values work reasonably well across datasets, so we did not use separate tuning for IoT23. The corresponding results are presented in Table 11 in the Appendix A.9, and we observe that IoT23 exhibits behavior similar to CICAndMal2017. To further clarify the effect of the distillation terms, we also report the case (α = 0, β = 0). This shows that while α and β can be empirically tuned, the model still performs reasonably well even without distillation. In practice, these weights mainly control the balance between past and current tasks, so they can be adjusted depending on the degree of task recency bias.
>
>
> **4. Backbone ablation for MLP, CNN, and ViT**
>
> To address your concern, we have added a backbone ablation study in the Appendix A.6. For fairness, all backbones are evaluated under the same settings to isolate the backbone from other effects: (i) 5-task CICAndMal2017 split, (ii) Fixed to tight buffer size = 6000, (iii) No refinement phase, (iv) Random selection.
>
> The result shows high stability of ViT. ViT maintains both higher accuracy and F1 score among three with a lower forgetting score. This suggests that ViT’s global attention better fits the malware traffic dataset than CNN or MLP. CNN achieved the highest accuracy in the early task, but exhibited the largest variance, with a rapid performance drop. Also, Conv1D learns patterns locally but generalizes poorly when the next task arises. In MLP, consistent lower performance, along with a wide gap between F1 and accuracy, indicates that its shallow architecture cannot capture the complexity of traffic embeddings.
>
> Overall, this ablation study supports adopting ViT as a backbone; even when other techniques are used, ViT still yields the most stable performance on malware traffic.
>
> **5. Torri and Trojan**
>
> Torri and Trojan are excluded because they contain fewer than 30 usable samples. In our analysis, classes with fewer than ~100 samples produced highly unstable embeddings, leading to inconsistent results across runs. To ensure that the evaluation remained reliable and reproducible, we restricted the experiments to classes with sufficient sample size, resulting in 9 classes for IoT23.
>
>
> **6. Re-encode/re-cluster**
>
> We used last-in first-out (LIFO) policy for each class: when the memory is full, the most recently added examples are removed.
>
> The model only selects the exemplar one time and does not update the buffer. Because more than half of the clusters used as examples contain three or fewer samples. This means that examples are drawn from many small, diverse clusters, so even after sequentially removing the most recent examples, the remaining set still covers the embedding space. In practice, we found this sufficient to preserve class diversity without re-clustering.

---

### Official Review · Reviewer_BCzy · 2025-11-01

**Soundness:** 3
**Presentation:** 3
**Contribution:** 3
**Rating:** 6
**Confidence:** 5

**Summary:**

The paper proposes TraMEL (Traffic-based Malware Exemplar Learning), a replay-based continual learning (CL) framework tailored to malware traffic analysis (MTA) with encrypted flows.

TraMEL combines imbalance-aware exemplar selection (random, class-mean, and diversity-oriented K-means) and a post-task exemplar refinement phase that reduces task-recency bias via a composite loss balancing cross-entropy with two distillation terms that anchor both past and current behavior. The authors define a refinement objective that explicitly trades off plasticity vs. stability.

Evaluated in Class-IL and temporal split settings on CICAndMal2017 and IoT23, TraMEL consistently outperforms strong replay baselines (ER, iCaRL, TAMiL), narrowing the gap to joint training while operating under strict memory budgets; the study also analyzes buffer size, refinement epochs, and loss weights.

**Strengths:**

- The authors focused the study on encrypted malware traffic at family level, a more realistic and fine-grained setting than conventional binary IDS settings.

- Two-stage design (diversity-aware exemplar selection + refinement) targeted to long-tailed, evolving families under tight buffers; the dual-distillation refinement is thoughtfully motivated.

- Comprehensive evaluation on CICAndMal2017 and IoT23 across Class-IL and temporal splits; repeated runs; reports task-wise/mean accuracy and forgetting.

- Demonstrates that memory-efficient replay with careful selection and refinement can deliver robust, fine-grained malware family classification under drift—relevant to practical SOC/defense pipelines with retention constraints.

**Weaknesses:**

- The paper argues that real deployments exhibit family recurrence, yet both Class-IL and temporal settings keep disjoint families across tasks (a conservative lower bound). This complicates claims about performance under true recurrence, open-set families, or re-emergence. A small synthetic recurrence experiment (e.g., re-introducing early families later) would strengthen the validity.

- Baselines are standard replay methods; however, related class-imbalance-aware CL or compressed/coreset replay variants are not included empirically. Even a small-scale comparison (or discussion) would contextualize TraMEL’s diversity-aware selection.

- Since the study emphasizes practical constraints and evolving distributions, it would be useful to discuss continual semi-supervised one-class approaches for malware detection on traffic—an adjacent but distinct framing that reduces label burden (e.g., Continual Semi-Supervised Malware Detection, MAKE 2024), and clarify how TraMEL’s supervised replay compares or could be hybridized.

**Questions:**

- Could you report a recurrence experiment (e.g., re-introduce a subset of Task-1/2 families at Task-5/6) to quantify retention when families return—potentially with smaller buffers? This would align the setup with the paper’s motivation (re-emergent variants).

- You note that larger k (e.g., >100) improves coverage. How sensitive are results to k vs. per-class sample counts, especially for extreme long tails? Any heuristics you recommend?

- Have you thought about adding automatic tuning (or an early-stopping criteria during refinement) to bound extra compute?

- Given labeling constraints in MTA, how does TraMEL interact with semi-supervised or one-class continual regimes? A discussion (or small pilot) contrasting TraMEL’s supervised replay with continual semi-supervised detection on traffic data would be valuable; for instance, see Continual Semi-Supervised Malware Detection (MAKE 2024).

- I suggest adding a compact pseudocode box for the three-phase loop. Please, also, include the seeds and data splits you used for temporal grouping for easier replication.

---

> ### Author Response · Authors · 2025-12-02
> **Rebuttal Response to Reviewer BCzy**
>
> Thank you for your valuable feedback. We address your concerns as follows:
>
>
> **1. Recurrence Experiment**
>
> By following your suggestion, we have conducted the recurrence experiment by splitting the Task 1 and 2 families (Beanbot, Nandrobox) into two subsets (90%, 10%) and trained the smaller one in the Task 4 and 6. This setting simulates the re-emergence of previously seen families in later tasks while keeping the buffer size fixed. The results (Table 7 and Figure 8 in the Appendix) show that the ViT encoder retains stable representations: performance on the recurrent portion remains high, even when the re-emerged samples are small.
>
> **2. Sensitivity of K**
>
>   - It is not particularly sensitive to the choice of K, even for long-tailed classes under the ViT classifier. In our analysis, we observed that roughly half of the clusters contained fewer than three samples, and more than 100 clusters were empty. Despite this fragmentation, the dispersed samples selected from these small clusters still retained stable representations.
>
>  - As a heuristic method, we recommend using intra-class variance. After training the feature extractor, calculate the intra-class variance. Using TraMEL-K, the mean for CICAndMal2017 is 0.4, and for IoT23, 0.08. In the paper, the best-performing K are 800 (CICAndMal2017) and 600 (IoT23), respectively, indicating that CICAndMal2017 is more dispersed than IoT23, which makes the use of larger K reasonable.
>
> **3. Early stopping**
>
> We have indeed considered early stopping or automatic tuning during the refinement phase, but given the computational cost, it is not necessary in our setting. As shown in Figure 7 in the appendix, the computational overhead of TraMEL increases only marginally even when the buffer size grows. However, to maintain greater accuracy, early stopping or automatic tuning could be a reasonable option to improve performance.
>
> **4. Semi-supervised setting**
>
> Extending TraMEL’s settings to semi-supervised learning is an interesting direction. In this work, we focused on supervised learning because most detection systems are based on supervised algorithms, and our baselines are replay-based continual learning. However, exploring how TraMEL’s supervised replay could be hybridized with semi-supervised objectives would be worth considering in the future work. We have included a discussion in the revised version.
>
> **5. Pseudo code, seed, and temporal grouping**
>  - We have added pseudo code in Algorithm 1.
>  - Mainly used seeds are 83,93,103,113,123. We have included this detail in the ``Reproducibility statement”.
>  - Temporal grouping (Task 7) is also added in Table 12 in the Appendix.
>
>
> **6. Coreset replay**
>
> We agree that coreset-based replay is a well-studied method in CL. We have considered performing a small scale experiment for the revised version, but could not do so due to the replicability challenge of the codebase of the prior work. However, we note that TraMEL also focuses on improving exemplar heterogeneity within a fixed budget, comparing it with a coreset-style exemplar selection method would provide additional context. We have added a discussion clarifying the difference between coreset selection and our approach in the revised version.

---

### Meta-Review · Area_Chair_vLmE · 2025-12-22

**Summary:**

This paper proposes TraMEL, an exemplar–based continual learning framework for malicious family identification in encrypted network traffic, aiming to address simultaneously: (i) the long-tailed sparsity of real traffic features, (ii) forgetting caused by task recency bias, and (iii) strict memory budgets. The method consists of two main components: (1) heuristic/clustering-based exemplar selection to balance class coverage and intra-class diversity; and (2) a buffer-only exemplar refinement stage after each incremental phase, coupled with a distillation loss, to mitigate forgetting and reduce recency bias. The paper reports improvements over iCaRL/ER/TAMiL on CICAndMal2017 and IoT23, and also investigates the impact of different model architectures.

Reviewers generally acknowledge the practical significance and research value of the problem. The overall structure is relatively concise, leveraging exemplar selection and buffer-only refinement to alleviate forgetting and recency bias. However, multiple reviewers raise critical concerns about the novelty of the methodological contribution, the adequacy of the experimental protocol and baseline comparisons, and the alignment between the experiments and the paper’s claims about “real-world open environments.” The authors provide additional analyses and clarifications in the rebuttal (e.g., on imbalance/recurrence, partial ablations, and protocol explanations), but the articulation of the core contribution and the empirical evidence remain insufficient to substantiate the claimed deployment constraints/open-world setting. Therefore, I currently lean toward rejection.

To reach an acceptable standard, the authors may consider: (i) more clearly positioning the contribution relative to existing replay/refinement methods and providing a mechanism-level explanation of why/when it works; (ii) adding key baselines/ablations to rule out confounding effects from the backbone and hyperparameter tuning; and (iii) systematically validating the open-world claims with settings and metrics that better match open-world scenarios (e.g., a unified protocol covering unknown classes, semi-supervision, and recurrence).

**Reviewer Concerns:**

**A. Points that are partially addressed/mitigated in the rebuttal**

1. The authors add settings and results for “family recurrence / non-mutually-exclusive classes,” partially addressing concerns about recurrence in realistic scenarios. However, this addition still largely stays within supervised settings with known labels, which is not sufficient to support broader claims about more general open environments.
2. The authors add analyses of exemplar selection strategies and distillation/refinement-related hyperparameters, and report performance variations across random/centroid/KMeans selection as well as different buffer sizes, partially alleviating concerns that the method relies on extensive hyperparameter stacking/tuning.
3. Disclosure of limitations: the authors acknowledge that refinement may sacrifice current-task accuracy and that a fixed buffer limits scalability, and they outline potential future directions.
4. Presentation: some terminology and presentation are improved, which helps readability.

**B. Remaining concerns**

1. **Weak novelty of the methodological contribution**: TraMEL is essentially a compositional design of exemplar selection plus refinement/distillation. The method-level insight (how it fundamentally differs from existing replay/refinement approaches) is insufficiently articulated. While the approach may be effective in practice, its methodological novelty for continual learning remains unclear. In other words, these strategies could potentially be further tailored and optimized for the specific encrypted-traffic malicious family identification setting, which would make the incremental contribution closer to a domain-specific adaptation of existing techniques.
2. **Experimental protocol and baseline coverage are still insufficient to support strong conclusions**: multiple reviewers question the breadth of baselines and whether some claims about deployment constraints/open-world settings remain largely narrative rather than empirically substantiated.
3. **Still weak evidence for “real open world / recurrence / unknown classes”**: the motivation emphasizes recurrence and openness in real environments, yet the main experiments are still primarily supervised closed-set continual learning. Although the authors add recurrence scenarios, there is still a lack of systematic settings, metrics, and sufficient empirical evidence to close the loop for more realistic conditions such as unknown classes and semi-supervised regimes.
4. **Scalability and deployability remain questionable**: the paper emphasizes tight memory budgets and deployment constraints, but the method relies on a fixed buffer as the number of tasks grows and introduces additional refinement training overhead; the authors also acknowledge limited scalability. This makes the evidence for “deployment-oriented” claims insufficient, and the practical boundary conditions should be more clearly defined and quantitatively evaluated.

**Reviewer Scores:**

Based on the overall reviews and the rebuttal, I do not expect substantial score changes. While the authors respond to some concerns and add supplemental analyses/experiments, the negative feedback concentrates on issues that are not fully resolved in the rebuttal, i.e., the novelty of the contribution, the experimental design and evidence chain, and the adequacy of baseline comparisons.

---

### Decision · Program_Chairs · 2026-01-26

Reject